# Summer Is Coming! Tackling Ocean Warming in Atlantic Salmon Cage Farming

**DOI:** 10.3390/ani11061800

**Published:** 2021-06-16

**Authors:** Ricardo Calado, Vasco C. Mota, Diana Madeira, Miguel C. Leal

**Affiliations:** 1ECOMARE, CESAM—Centre for Environmental and Marine Studies, Department of Biology, University of Aveiro, Campus Universitário de Santiago, 3810-193 Aveiro, Portugal; d.madeira@ua.pt (D.M.); miguelcleal@ua.pt (M.C.L.); 2Nofima AS, P.O. Box 6122, NO-9291 Tromsø, Norway; Vasco.Mota@nofima.no; 3UCIBIO—Applied Molecular Biosciences Unit, Faculdade de Ciências e Tecnologia, Universidade NOVA de Lisboa, Quinta da Torre, 2829-516 Caparica, Portugal

**Keywords:** climate change, heat stress, phenotypic plasticity, *Salmo salar*, thermal tolerance

## Abstract

**Simple Summary:**

Atlantic salmon (*Salmo salar*) has become a commodity worldwide. The culture of Atlantic salmon is by far the most well-developed branch of marine finfish aquaculture, with this species ranking among the top ten most highly produced in global aquaculture. While Atlantic salmon has been commonly farmed in sea cages located in colder waters (e.g., in Norway, Chile and Tasmania), these regions can experience the negative impacts of heat waves that push seawater temperature above values tolerated by this species. These climate-change-driven shifts in water temperature can be associated with mass mortality events and urgent actions are needed to cope with a changing ocean. This paper reviews the thermal limits of adult Atlantic salmon and lists the negative effects driven by heat stress. We highlight how biotechnology and the genetic diversity of wild populations may help producers to tackle this challenge. Selective breeding programs and other more advanced biotechnological solutions (e.g., gene editing) may play a key role in this quest to produce new strains of Atlantic salmon that more readily tolerate higher water temperatures, without compromising productivity and profitability.

**Abstract:**

Atlantic salmon (*Salmo salar*) cage farming has traditionally been located at higher latitudes where cold seawater temperatures favor this practice. However, these regions can be impacted by ocean warming and heat waves that push seawater temperature beyond the thermo-tolerance limits of this species. As more mass mortality events are reported every year due to abnormal sea temperatures, the Atlantic salmon cage aquaculture industry acknowledges the need to adapt to a changing ocean. This paper reviews adult Atlantic salmon thermal tolerance limits, as well as the deleterious eco-physiological consequences of heat stress, with emphasis on how it negatively affects sea cage aquaculture production cycles. Biotechnological solutions targeting the phenotypic plasticity of Atlantic salmon and its genetic diversity, particularly that of its southernmost populations at the limit of its natural zoogeographic distribution, are discussed. Some of these solutions include selective breeding programs, which may play a key role in this quest for a more thermo-tolerant strain of Atlantic salmon that may help the cage aquaculture industry to adapt to climate uncertainties more rapidly, without compromising profitability. Omics technologies and precision breeding, along with cryopreservation breakthroughs, are also part of the available toolbox that includes other solutions that can allow cage farmers to continue to produce Atlantic salmon in the warmer waters of the oceans of tomorrow.

## 1. Introduction

Atlantic salmon (*Salmo salar*) ranks in the top 10 of the most produced and most valuable marine fish species in world aquaculture [1]. Although global production volume has been relatively stable, at around 2 million metric tons (MMT), salmon price has displayed some volatility over the past 10 years, mostly due to reduced short-run elasticity of supply [2]. Disease outbreaks and parasites still pose a significant threat to the supply chain, as well as to the sustainability and profitability of sea cage farm operations [3]. The threat posed by sea lice (*Lepeophtheirus salmonis* and *Caligus elongatus*) has long been recognized [4] and continues to be of particular concern, as these parasitic copepods continue to induce high levels of mortality that result in serious production shortfalls [3]. Another factor that also threatens cage farming operations of Atlantic salmon is the occurrence of unsuitable seawater temperature, likely a consequence of ongoing climate change. While losses promoted by cold are certainly of concern and may have even been somehow neglected by the scientific community [5], the present work will only focus on the impact of warming seawater temperature. Higher seawater temperatures are known to be associated with disease outbreaks and poorer growth performances in Atlantic salmon, namely when thermal anomalies associated with marine heatwaves (MHW) occur during cage farming operations [6,7,8,9,10], as recently reported in Tasmania [10].

Salmon cage farming has traditionally been located at higher latitude regions, such as Chile, Canada, the Faroe Islands, Norway and Scotland (Figure 1). This spatial distribution is driven by existing environmental conditions in these locations, which are optimal for salmon cage farming at the sea surface, such as cold-water temperatures (8 to 14 °C), optimal biological conditions and sheltered coastlines [11].

However, cold-water regions are being increasingly impacted by climate change, with ocean warming and recurrent heat wave events being recorded in recent years [12]. This scenario has been reported at farming sites in Newfoundland (Canada), resulting in major mass mortality events in salmon cage farming operations, and has raised a global awareness among multiple stakeholders to address this issue [13,14]. However, a thorough report concluded that even though an increase in temperature was the likely major driver of salmon mortality, the combined impact of multiple stressors clearly played a role in worsening the situation [15]. A combination of heat stress, hypoxia, sea lice infestation and an algal bloom in the sea cage area were preponderant factors in these mortality events [15]. Still, the full impacts of increased sea surface temperature on the aquaculture industry have been difficult to quantify due to the lack of publicly available information and the challenge to distinguish natural variations from climate change [5]. While it is imperative to address global climate change, ocean warming and MHW events respond very slowly to mitigation actions. Alternative solutions are needed to deliver more immediate and tangible results in the prevention of mass mortality events associated with heat stress and secure a responsible and sustainable Atlantic salmon aquaculture industry.

The present review starts by covering the current knowledge on adult Atlantic salmon tolerance limits to higher seawater temperatures, as well as the eco-physiological consequences of heat stress and how it affects sea cage aquaculture production cycles of this species. After raising awareness towards this problem, innovative biotechnological solutions are pinpointed and discussed, with emphasis on thermo-tolerant strains as a potential solution to address the bottlenecks that the industry will face in upcoming years.

## 2. Adult Atlantic Salmon Thermal Tolerance and Physiological Responses to Heat Stress

Predicted increases in temperature are projected to have significant impacts on ectothermic organisms, such as fish, due to the thermal dependence of metabolic activity [16,17,18,19,20]. The Intergovernmental Panel on Climate Change (IPCC) reports an overall average increase of 0.85 °C in global temperature since 1880, with sea surface temperature rising by 0.8 °C [21]. Noticeably, the past three decades have been successively warmer, with a rate of temperature increase in global surface temperature of +0.2 °C per decade [22]. Climate projections for the next century show an increase ranging from 1 to 3 °C in sea surface temperature [23]. Additionally, global circulation models predict an increase in ocean heat content due to ice sheet and glacier mass loss and an increase in extreme events with regards to their frequency, intensity and duration (e.g., heat waves, cold spells, La Niña and El Niño). Overall warming trends have been more pronounced in the Northern than the Southern Hemisphere [24] and warming rates are generally greater at higher latitudes when compared to equatorial regions. In particular, MHW, defined as temperature anomalies where sea surface temperatures exceed the 90th percentile of the local long-term climatology for at least 5 consecutive days [25], can be particularly damaging for marine species [26], with potential impacts on aquaculture production (e.g., [27]). A study by Oliver et al. [12] reported that MHW intensity and annual MHW days are increasing worldwide, with significant, widespread and persistent effects on marine ecosystems. However, overall projected MHW intensity was strongly dependent on greenhouse gas emission scenarios. Still, compiling information provided by Oliver et al. [12], Smale et al. [26] and Viglione [28], the following trends could be depicted: (i) some of the most prominent MHW of the past decade happened in salmon farming areas, especially around Tasmania, the Gulf of Maine (USA), Newfoundland (Canada) and Iceland (see Figure 1 from Viglione [28] and Laufkötter et al. [29]), (ii) areas where salmon farming occurs show a projected increase between +1 to +2 MHW days per year per decade (Chile) and +4 to +5 MHW days per year per decade (ex. Tasmania) (see Figure 1 from Smale et al. [26]) and (iii) salmon production areas have historically been subjected to heatwaves of maximum intensity between +1 to +3 °C, with total annual MHW days between ~27 (e.g., Chile) to ~33 days (e.g., Norway, UK, Tasmania). In the RCP8.5 scenario, an increase in intensity (+1 to +4 °C, depending on the region) and MHW days (~80 to 320, depending on the region as well) is expected [12], calling for the attention of the aquaculture industry towards climate change adaptation in the future. For example, the annual cumulative intensity of MHW has increased 340% over the past 30 years in southern Norway (Flødevigen Research Station) [30], with unusually high water temperatures (>18 °C) being registered in the south of Norway in recent years [31]. In the Gulf of Maine, where the salmon aquaculture industry is also well established, surface waters have increased at a rate of +0.23 °C per year in the past decade [27]. Moreover, records of a persistent MHW over the Gulf of Maine in 2012 led to a peak of >18 °C in sea surface temperature [27]. Another MHW was registered in the area in 2016. Overall, climatic and oceanographic data indicate that the warm water season in the Gulf of Maine is, on average, 3 to 4 weeks longer in comparison with long-term climatology (see [27]).

Cold-water fish species in Northern regions, including Atlantic salmon, are especially vulnerable to climate change, particularly because of the detrimental effects that excessive heat promotes on their physiology (e.g., increased stress, decreased growth and reproductive success, mortality) [33,34,35]. Atlantic salmon is natively distributed along the North Atlantic in temperate, sub-Arctic and Arctic zones (Figure 2) and has been introduced into New Zealand, Chile, Argentina and Australia, depending on cold and clear water to thrive [36]. Still, average thermal regimes within the distributional range of Atlantic salmon can vary widely in time and space [37]. For instance, mean seasonal variation can be between −1 °C in the winter and 16–18 °C in the summer in Norway, Russia and Canada [38,39,40], while temperatures as high as 23 °C have been registered in Tasmania (Australia) [41]. Maximum temperatures of up to 20 °C have also been recorded in Newfoundland waters (Canada), as well as steep daily variations of 10 °C [42]. Atlantic salmon has been classified as highly vulnerable to climate change due to its predicted exposure to elevated temperature, high sensitivity to warming and dependence on environmental cues to successfully migrate to spawning grounds [43].

Temperature is a critical factor shaping salmon growth and survival [44,45] and can act as a selective force shaping key life-history events, such as timing of breeding, hatching and development [46]. Moreover, increased water temperatures are known to affect Atlantic salmon behavior [47], smoltification [48,49] and migration [50,51]. However, knowledge on upper thermal limits and the impacts of elevated temperature on Atlantic salmon is incomplete, given that most studies to date were carried out on early or juvenile life-stages. Long-term studies on this topic are scarce [52], with most available works addressing acute exposure to higher temperatures. The CTmax of larger salmon, assessed using a warming rate of 2 °C h^−1^, was 26.2 °C for 1132 g salmon [53] and 26.5 °C for 625 g salmon [54]. Further studies should use standardized methodologies to obtain comparable results. As CTmax trials are acute experiments that may not represent ecologically realistic conditions, a new type of experiment has been recently proposed, termed ITmax (incremental thermal challenge), in which temperature is increased at a rate that mimics natural conditions at sea (see [55]). If we consider the CTmax estimated by Leeuwis et al. [53] and Penney et al. [54] and the methodology employed by Gamperl et al. [52], in which salmon were exposed to a warming rate of 1 °C week^−1^, we conclude that the ITmax of Atlantic salmon (~22 °C) is approximately 4 °C lower than the CTmax (~26 °C). Still, Gamperl et al. [52] highlight that temperature increases beyond 18–20 °C can negatively impact salmon production.

The thermal range for growth of Atlantic salmon is between 4–6 to 19–22.5 °C, with an optimum at 13–16 °C [56,57,58,59,60] (but see [61]). Interestingly, the optimum temperature for growth coincides with the estimated optimum for aerobic scope (approximately 15 °C, based on Arrhenius breakpoint temperature) [62]. Such observations fall well within the oxygen- and capacity-limited thermal tolerance (OCLTT) theory, which states that the thermal window for optimal performance matches the aerobic scope and capacity to supply oxygen to tissues [63,64]. As temperature increases, there is a mismatch between oxygen demand and supply, ultimately defined by cardio-respiratory failure. Still, there is no consensus about the OCLTT hypothesis (e.g., [65,66]) including in salmon, as other studies have shown that the aerobic scope of Atlantic salmon (~450 g fish) remains the same across different temperatures (13, 18, 23 °C) [61]. Thus, thermal tolerance limits of fish could be oxygen-independent [67] and be related to other mechanisms, namely loss of macromolecular integrity (e.g., [68]), dysregulation of sarcolemmal ion regulation (altering ventricular excitation, e.g., [69]) and disruption of biological membranes and antioxidant capacity (e.g., [70], and see [71] for a brief discussion). Other authors have also shown that thermal tolerance limits are modulated by heat-induced neural dysfunction associated with limited oxygen availability [72]. Moreover, while acute exposure to 20 °C increased maximal mitochondrial respiration by 50% and reactive oxygen species (ROS) production by 60%, long-term acclimation (2 months) of Atlantic salmon to 20 °C allowed fish to maintain mitochondrial coupling and aerobic capacity, while reducing ROS production by 30% [73]. According to the same authors, it seems that mitochondrial function is sufficiently plastic to enable the survival of salmon at 20 °C for several weeks or even months. Phenotypic plasticity and acclimation capacity of Atlantic salmon is thought to be partially mediated by DNA methylation changes which modulate gene expression [74]. The same authors refer to the fact that exposure to a higher temperature (20 °C) has been shown to alter methylation of CpG sites (exposure-time-dependent) within genomic regulatory elements of genes involved in responses to temperature, oxidative stress, apoptosis and metabolism. In a complementary study, Beemelmanns et al. [75] showed that the liver transcriptome of Atlantic salmon undergoes significant changes upon exposure to thermal stress (12 to 20 °C, rate of 1 °C week^−1^, alone and in combination with hypoxia). Responses were similar between fish exposed solely to thermal stress or thermal stress and hypoxia, with an upregulation of genes involved in heat shock response, endoplasmic reticulum stress response, apoptosis and immune defense being observed in both groups, while genes involved in metabolic processes, proteolysis and oxidation–reduction were downregulated [75]. The authors also found that transcriptional changes were strongly correlated with impaired growth performance, thus highlighting the role of these genes in fish health. A proteomic study in Tasmanian pre-harvest salmon exposed to 21 °C for 43 days also showed significant liver proteomic remodeling at increased temperatures [76]. In particular, genes associated with oxidative stress pathways, the endoplasmic reticulum stress response and amino acid degradation were upregulated at high temperatures, whereas those associated with transcription and translation mechanisms, protein degradation and cytoskeletal elements were downregulated. The authors also report that proteomic remodeling was accompanied by a reduced condition factor and hepatosomatic index. It should be noted, however, that a thermal preference and tolerance shift is expected during ontogeny [77,78]. Usually, upper thermal tolerance limits in fish are expected to increase from eggs to juvenile stages and then decrease again in mature adults (Figure 3) [63,78]. Ultimately, stage-specific sensitivity (Appendix A) and regional variability in water temperature will be determinant factors shaping the potential impacts of global warming on Atlantic salmon aquaculture.

## 3. Impacts of Heat Stress on Atlantic Salmon Cage Farming

Different Atlantic salmon populations are already experiencing thermal stress in the wild and in farm cages [10,79]. While water temperature at salmon farms can range from −1 to 19–20 °C, a more efficient growth has been reported to occur between (i) 10 and 14 °C ([80]) and (ii) 13 to 17 °C [6]. In general, if temperature increases beyond 18–20 °C, specific growth rate and condition are negatively affected [52,81]. Accordingly, recent studies have shown that feed consumption of Atlantic salmon starts to decrease at 18–19 °C, with a rapid decline as temperature surpasses 20 °C [52]. Still, according to the same study, an incremental temperature increase from 12 to 20 °C at a rate of 1 °C week^−1^ did not lead to changes in specific growth rate, wet weight gain or condition factor, and did not induce mortality, suggesting that salmon can withstand up to 20 °C. In an experiment where 2 kg salmon were exposed to 14 and 19 °C for 56 days, fish held at 19 °C showed a reduction of ≥ 50% in daily feed intake, growth and feed utilization when compared to fish held at 14 °C [82]. A concomitant decrease in energy reserves and little retention of ingested fat was also detected in salmon held at 19 °C, associated with high maintenance costs at elevated temperature, also reported in the same study. Conversely, other studies report a slight increase in feed conversion ratio at warmer temperatures (~18 °C) due to higher metabolic rates and energy losses in feces and excretory products (see [52]). When temperature surpasses 20 °C, appetite, specific growth rate, condition factor and hepatosomatic index decrease in Canadian and Norwegian Atlantic salmon [52,61]. Swimming activity is also influenced by temperature, as shown by Hvas et al. [61]. In that same study, 450 g salmon were acclimated to different temperatures (3, 8, 18, 23 °C) for 4 weeks, and critical swimming speed peaked at 18 °C and decreased significantly at both extreme cold (3 °C) and heat (23 °C).

Heat stress and warming temperatures are known to negatively impact salmon performance, as higher water temperatures promote lower oxygen solubility. Increased water temperature and hypoxia co-occur at salmon cage sites, especially during the summer season (see [41,83]). Dissolved oxygen levels have a major impact on fish performance traits, such as feed intake, feed conversion ratio (FCR) and fish growth. For instance, extreme temporal and spatial variation in DO distribution within farming cages leads to salmon experiencing suboptimal conditions, which negatively impact growth performance [84]. Further studies showed that, even if fish are held at optimal temperature (17 °C), moderate hypoxia leads to a significant decrease of about 25% in feed consumption and growth, with a concomitant increase in mortality [85]. Although adult salmon show a preference for temperatures around 16 to 18 °C [86], selection of preferred temperatures is constrained by their active avoidance of low DO (<35% saturation) at the bottom of the cage. In addition to low DO, Atlantic salmon also avoid warm surface waters (>20.1 °C), which leads to a considerable contraction in available vertical habitat when farmed in surface cages. Ultimately, and despite their avoidance behavior, fish may well end up spending a considerable amount of time in waters with suboptimal DO (<60% saturation) [41].

The physiological and performance impacts of warming water and heat stress on Atlantic salmon ultimately lead to mortality events, which seriously jeopardize the viability of cage farm operations. Thus, the occurrence of MHW can lead to devastating effects on animal welfare, performance and early maturation, as well as mortality of Atlantic salmon farmed in cages [10,51,77,87,88,89,90,91,92,93]. Ultimately, stress responses at molecular and whole-organism level, changes in growth performance, mortality events and overall thermal tolerance may depend on several factors, including the rate of temperature increase and exposure duration (e.g., [52,94]). Laboratory studies on Atlantic salmon have shown that a short-term exposure (a few weeks) to elevated temperatures (>18 °C) may have a minimal effect on salmon. However, prolonged exposure (>1 month) can induce detrimental effects, such as a substantial decrease in specific growth rate, thus impacting production in Atlantic salmon ranging from 400 to 800 g [52]. This information is particularly relevant considering recent MHW trends. For example, the austral summer heatwave of 2015/2016 in Tasmania pushed seawater temperatures to values over 18 °C for 117 days. During this heatwave, water temperature went over 20 °C for 83 days (peaking at ~23 °C) in areas where Atlantic salmon farming cages were deployed. These extreme conditions led to liver damage, a cessation of feed intake, development of anorexia and loss of flesh coloration with potential implications for the perceived quality of the fillet and, consequently, market price [10]. Hence, the impacts of chronic temperature stress need to be carefully addressed by the salmon aquaculture industry. Recent newspaper editions have highlighted numerous salmon mortality outbreaks due to elevated temperature, emphasizing the need for climate change adaptation in the aquaculture industry [14,95]. Particularly, Canadian salmon farms experienced high temperatures and low oxygen conditions, which resulted in fish mortalities [15]. Sustained high water temperatures resulted in the death of ~2.6 million Atlantic salmon between late August and early September at 10 Northern harvest farms in southern Newfoundland, causing losses estimated at USD 5.5 million [96]. Similar issues have been recorded in Norway, where higher temperatures were also thought to be the key drivers of a large algae outbreak that resulted in the mortality of several million Atlantic salmon in the northern part of the country, and economic losses of more than USD 82 million [97]. Similarly, a recent study reports that climate change threats to salmon farming in Chile include significant changes in water temperature, and consequent declines in dissolved oxygen, along with the occurrence of harmful algal blooms and diseases [98]. A combination of drier conditions and El Niño events is predicted to occur in the Patagonian region, resulting in extreme dry summers [99]. Such conditions favor the development of harmful algal blooms, such as the one observed in 2016, which killed 12% of all Chilean salmon production. The key drivers of this algal bloom were related to an increase in sea surface temperature and a reduction in precipitation and freshwater inflow to the fjords, due to an El Niño event superimposed on a positive phase of the Southern Annular Mode [98,99]. These scenarios should be carefully considered by the salmon farming industry, especially given the increasing trends of MHW worldwide (e.g., [100]). Accordingly, modeling studies had already predicted that farms in the south of Norway (e.g., at Lista) would potentially experience a negative effect on productivity due to an increased probability of disease outbreaks and algal blooms, impairing growth and inducing fish mortality [101,102,103]. Still, Atlantic salmon seems to have the capacity to sustain innate immune system responses even under high temperatures (20 °C) and moderate hypoxia (65–75% air saturation) [104] which could help fish to face pathogens. In contrast to the south of Norway, an increase in average sea surface temperature due to climate change could enhance salmon farm productivity in northern areas of this country (e.g., at Skrova), by enhancing salmon growth [62]. Still, geographic differences call for the importance of calibrating climate change projections to local conditions at farm sites, a key aspect to inform and develop farm-specific adaptation strategies [105].

Temperature-driven stress is not a recent problem for Atlantic salmon. Indeed, mortality was described in wild salmon stocks as early as the 1940s, when individuals were exposed to temperatures of approximately 29.5 °C in Moser River (Nova Scotia, Canada) [106]. Temperature-related problems are becoming more frequent due to ongoing climate change, and are synergistically potentiated by other issues driven by cage-farming operations. For instance, novel cage technological solutions for pathogen control, such as “lice skirts”, provide a physical barrier around the sea cage to prevent sea lice, but have the indirect effect of reducing water flow. This reduced water flow results in low oxygen saturation during summer months when water temperature is higher, which resulted in decreased animal welfare and growth performance in Norwegian farms [107]. Overall, it is now unequivocal that heat stress promoted by high water temperature during grow-out of Atlantic salmon in sea cages is already negatively impacting the farming industry, which urgently needs to adapt to this new reality.

## 4. Accessing Local Adaptation and Phenotypic Plasticity to Foster Temperature-Dependent Selection of Atlantic Salmon

Negative effects of ocean warming in cold-water fish may be buffered by phenotypic plasticity or adaptive evolution. While phenotypic plasticity implies that a genotype can produce a range of phenotypes along an environmental gradient [108], adaptive evolution implies that heritable variation created by mutation is acted upon by natural selection, changing allele and trait frequencies in a population throughout time [109,110]. Phenotypic plasticity is well-established for salmonid thermal windows, as these shift according to acclimation temperature and recent thermal history [111,112,113]. However, evidence of local adaptation in Atlantic salmon is mostly circumstantial, being based on ecological correlates in fitness-related traits and translocation outcomes [114]. While patterns of neutral genetic variation are well-studied, adaptive variation in non-neutral genes is under-represented in the literature [114]. In theory, thermal performance could be expected to vary between populations, as (i) the thermal regime varies across latitude in a predictable way, (ii) salmon display reproductive isolation due to their homing behavior, potentially reducing gene flow among populations and leading to local adaptation and (iii) most phenotypic traits have a genetic heritable component (including thermal tolerance in fish [115]) and variation in these traits leads to fitness differences [114,116]. However, some studies suggest that phenotypic plasticity has been selected for in Atlantic salmon, instead of thermal adaptation [117]. Cases that support phenotypic plasticity over adaptation include a comparison of Norwegian populations of Atlantic salmon, which showed similar optimal temperatures for growth despite differences in thermal regimes [118]. In contrast, similar cardiac plastic responses (a proxy of thermal tolerance) to high temperatures were recorded between wild Atlantic salmon strains from northern and southern Norway [38]. While this observation might be associated with an absence of a local adaptation of the different studied populations, the genetic divergence between these populations might not be enough to detect differences in thermal tolerance physiological traits. Notwithstanding, there is some evidence of adaptive divergence in thermal performance-related traits in Atlantic salmon and other salmonids [119,120,121,122]. Northern populations of Atlantic salmon display a faster growth during summer and autumn when compared to southern conspecifics, growing faster during winter and spring [123]. This pattern is explained by counter-gradient variation in digestive performance [124]. Similarly, a selection for increased growth efficiency under low water temperatures in Atlantic salmon has also been documented [119]. A clinal variation in immune function, in relation to temperature, has been reported for Atlantic salmon, with major histocompatibility complex (MHC) diversity being higher in populations from warmer habitats; this finding likely reflects the ability of these fish to better cope with the greater pathogen diversity, infectivity and virulence associated with higher temperatures [125]. Thus, such populations may serve as gene bank candidates for aquaculture development projects under climate change scenarios, when aiming to improve fish immune function and reduce disease-outbreak-associated mortality.

Temperature-dependent selection has also been proposed to explain clinal variation in the malic enzyme 2 locus in Atlantic salmon, as populations from warmer habitats have higher frequencies of the MEP-2∗100 allele, while populations from colder habitats have higher frequencies of the MEP-2∗125 allele [126,127]. In fact, malic enzyme 2 variation has been associated with differences in growth rate, survival and male maturation [128]. Allelic frequencies for this gene and other protein-coding loci were evaluated in domesticated and wild strains of Atlantic salmon, showing that MEP-2∗100 allele displayed a higher frequency in DOM-1 (0.914), DOM-5.95 (0.639) and DOM-2 (0.569) than the wild strain from River Namsen in Norway (0.525) [129]. Temperature-dependent selection has also been proposed to explain allelic variation in genes related to anaerobic and aerobic metabolism in other salmonids, namely (i) lactate dehydrogenase gene among Asian populations of sockeye (*Oncorhynchus nerka*) and pink salmon (*Oncorhynchus gorbuscha*) and the British Isles’ brown trout (*Salmo trutta*), and (iii) isocitrate dehydrogenase locus in steelhead trout (see Taylor [119] for a review). Nevertheless, links between such allelic variation in strains/populations and adaptive variation in thermal tolerance remain elusive and need to be clearly established. Only then, protocols can be developed by the aquaculture industry to foster potential adaptation strategies.

Other studies carried out with farmed fish showed variation in temperature tolerance among Atlantic salmon families produced from a group of 70 parental fish coming from a broodstock at Cooke Aquaculture (Blacks Harbour, NB, Canada). Higher thermal tolerance was associated with increased hypoxia tolerance, larger ventricle mass and higher myoglobin levels (which facilitate oxygen supply and transfer to tissues), thus supporting the OCLTT model [115]. Such results suggest that adequate breeding designs and genetic selection of high performers in aquaculture facilities could be an efficient strategy to develop more thermo-tolerant stocks that can better cope with climate change, as suggested by recent studies [10]. In fact, assisted evolution through selective breeding, assisted gene flow, hybridization, conditioning, transgenerational acclimation and epigenetic programming have been proposed as major tools in conservation efforts under climate change, as means to accelerate natural evolutionary processes to enhance certain traits, such as stress tolerance [130,131]. Changes in rearing temperature during development may also improve the thermal tolerance and performance of fish, taking advantage of developmental plasticity (or irreversible non-genetic adaptation), a process whereby environmental conditions during early development affect the phenotypes of subsequent life stages [132,133,134]. While some of these methods are already well-established in the aquaculture industry to enhance growth and disease resistance, others, such as epigenetic programming and short stress exposure, can be further developed, as already suggested [135,136]. Beemelmanns et al. [74] demonstrated that at a water temperature of 20 °C, Atlantic salmon display a shift in methylation of cytosine-phosphateguanine (CpG) dinucleotides in five biomarker genes (cirbp, serpinh1, prdx6, ucp2 and jund), with these also being correlated with gene expression. Another important finding in this study was that shifts in CpG methylation were shaped by the duration of thermal stress, promoting either reversible or persistent effects. Time-mediated dynamics between stress-induced shifts in CpG methylation and gene expression can be a pivotal epigenetic mechanism that allows Atlantic salmon to better acclimatize with changing baseline conditions in seawater temperature, as currently foreseen to occur [23]. It may also be worth investigating how variable levels of micronutrients (e.g., vitamins and minerals) in formulated aquafeeds can promote epigenetic regulation in a dose-dependent manner. While the sole available study to date on this topic specifically monitored gene expression in the lipid metabolism of Atlantic salmon parr and smolts [137], it will be certainly worth exploring if epigenetic regulation of biomarker genes associated with thermal tolerance may also be achieved by adjusting micronutrients incorporation in aquafeeds. Ultimately, the development of these approaches within a context of climate change mitigation is paramount to foster aquaculture production over the next decades.

Northwestern Portugal harbors the southernmost populations of *Salmo salar* native to the eastern Atlantic [138]. However, the construction of damns, along with overfishing and habitat destruction (namely spawning grounds and habitat refugia for alevins) have pushed these populations to the brink of extinction [139]. While the IUCN Red List of Threatened Species ranks Atlantic salmon as a “Least Concern” (LC) species, at a global scale [140], in Portugal this species is ranked as “Critically Endangered” (CR) [141]. Indeed, the captures reported for Atlantic salmon by Portuguese authorities in the year 2000 in the transboundary Minho river (separating Portugal and Spain) were of solely five specimens [139], and since then no change to this scenario has been reported. If ongoing global climate change maintains its trend, northern areas of Europe may well represent a refugium for Atlantic salmon genetic diversity in the future [142]. Being a cold-water species, Atlantic salmon in the southernmost limit of its natural range will likely be more vulnerable to extinction due to climate change [143]. A latitudinal cline of genetic variation, with higher values in northern areas and lower values in southern areas, has already been reported for Atlantic salmon [142]. Population declines have been reported to be consistently temperature associated and impacting genomic regions related to metabolic, developmental and physiological processes [144]. While moderate climate change scenarios, coupled with the preservation of wild populations and their habitats, may still allow some southern populations of Atlantic salmon to adapt and thrive [145], the persistence of extreme climate shifts, poor river flow management and habitat loss will certainly result in the local extinction of Portuguese populations. Farmed and wild salmon differ in a number of traits (e.g., molecular–genetic polymorphisms, growth, physiology and gene transcription), with the offspring of farmed specimens not being locally adapted and displaying a lower lifetime fitness than wild conspecifics [146,147]. As already reported for populations in northern Spain [148], any restocking efforts on these dwindling populations of Atlantic salmon in Portuguese rivers will be of little value if they do not account for the negative effects associated with introgression promoted by using non-local gene pools. Ultimately, using non-local gene pools will impact life-history traits and end up decreasing the resilience of these populations. Indeed, the effects of genetic drift caused by the significant decrease in effective population size may soon outweigh those of selection [144] and push these Atlantic salmon populations beyond a point of no return.

The present scenario frames a complex conundrum for the conservation of the southern populations of Atlantic salmon: how do we raise public awareness and advocate for the conservation of a species which is globally perceived as not being threatened? The occurrence of genetic erosion and loss of genetic diversity have already been flagged as direct consequences of climate change [149]. The loss of cryptic genetic diversity can result in the irreversible loss of adaptive potential to less favorable environmental conditions [150]. For the aquaculture industry, the loss of the genetic diversity of the southernmost populations of *Salmo salar* may be a serious setback, as it can compromise the chance to selectively breed lineages which have been naturally selected to better endure a warmer climate. It is therefore urgent to take action and safeguard the screening and stocking of the genetic diversity of these Atlantic salmon populations before it is too late.

## 5. The Role of the Salmon Breeding Industry

Selective breeding programs continue to have a major positive impact in aquaculture [151,152,153], with Atlantic salmon being no exception. The positive impact of selective breeding programs on the growth performance of this species is unquestionable when cultured specimens are compared with wild conspecifics. Glover et al. [154] reported an increase of up to 131% in body weight at harvest. However, unlike the enhancement of growth performance, processing yield or disease resistance, the tolerance of Atlantic salmon to higher water temperatures through selective breeding has only recently started to be addressed by the industry [155]. According to Sae-Lim et al. [156], selective breeding programs should target fish robustness, with “robust” fish being defined as specimens that are able to thrive under a broader environmental scope during production. According to these authors, “environmental sensitivity” may soon emerge as a new trait being favored by Atlantic salmon selective breeding programs. As already advocated for in other agro-sectors [157], Atlantic salmon aquaculture will also have to rapidly adapt to climate uncertainties to maintain profitability. Unlike most other aquaculture activities worldwide, selective breeding programs are well-established in Atlantic salmon farming [151]. Therefore, this specific branch of finfish aquaculture may more rapidly put into practice the fine tuning of such breeding programs.

In a warming ocean with MHW events becoming recurrently more frequent, understanding the mechanisms ruling transgenerational epigenetic inheritance can be a game changer for the Atlantic salmon breeding industry. Indeed, multiple studies addressing this topic, from an ecological and integrative perspective, highlight that species survival in a changing ocean will largely rely on their ability to tune their phenotype to a warmer environment over consecutive generations (a process termed transgenerational acclimatization) [158,159,160]. This “epigenetic buffering” at the population level may allow species to adjust their phenotype and cope with immediate impacts promoted by a warming ocean, thus allowing them “to buy time” for genetic adaptation to occur [161]. The Atlantic salmon industry must enhance its research efforts to screen the methylome of Atlantic salmon under acute and chronic thermal stress, within and over multiple generations, to generate life-stage-specific DNA methylation libraries for the species. Such databases can allow researchers to identify permanent epigenetic markers that may better predict which batches of fish would be more thermo-tolerant and, consequently, have a lower probability of being negatively impacted by heatwaves [74,75].

Overall, by using biotechnological tools currently available, enhanced breeding programs can allow farmers to better cope with farm-specific and regional-scale environmental constraints triggered by climate change.

## 6. The Way Forward

Genetic engineering and omics technologies will likely play a key role in the adaptation of Atlantic salmon aquaculture to climate uncertainty. However, as already highlighted by Myhr and Dalmo [162], several ecological, legal and ethical issues may arise. Gene-editing technology employing CRISPR/Cas9 allows for precision breeding, an approach that encloses a remarkable potential to improve aquaculture worldwide [163]. As discussed by Yang et al. [164], “precision breeding of individuals can be achieved by combining population genomic information at multi-omics levels together with genomic selection and genome editing techniques”. As such, desirable traits being targeted by precision breeding can be more efficiently transferred between conspecifics without the constraints of genetic dilution that occur when employing conventional breeding approaches [163]. Ongoing breakthroughs on the cryopreservation of Atlantic salmon gametes [165] can safeguard the long-term storage of valuable genetic resources, an urgent need transversal to the whole production of aquatic animals [166]. If the genetic diversity of the southernmost populations of Atlantic salmon is not lost and is rapidly unraveled, it may contribute to paving the way towards the development of a “warm water tolerant” strain through precision breeding. By taking advantage of the already existing high-quality assembly and annotation of the Atlantic salmon genome [167], one can more efficiently explore the natural genetic diversity of the southernmost populations of this species, thus favoring the selection of specific genomes and phenotypes and avoiding the use of non-species-specific genetic material.

## Figures and Tables

**Figure 1 animals-11-01800-f001:**
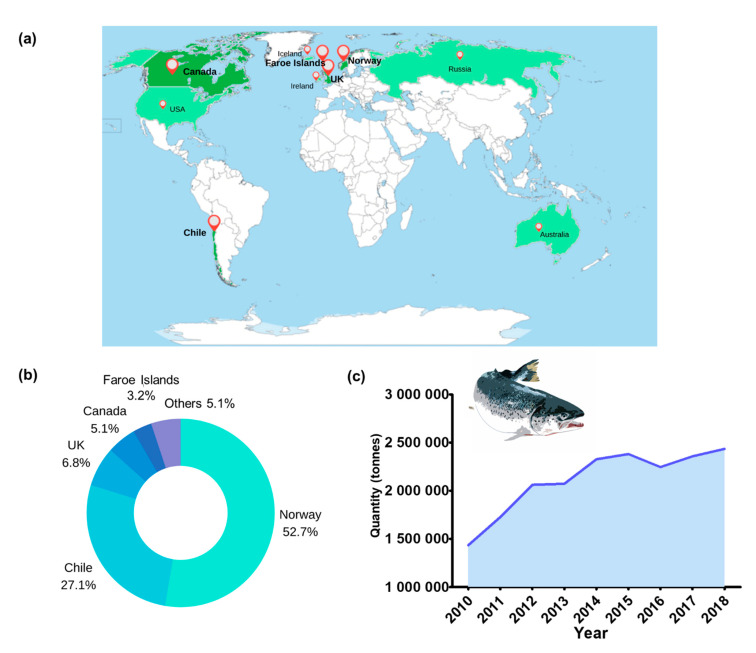
Aquaculture of Atlantic salmon (*Salmo salar*). (**a**) Top 10 producer countries, with top five being colored in dark green and highlighted in bold [1,2,32]; (**b**) production (in percentage) of top 5 countries [1]; and (**c**) global production (in tons) from 2010 to 2018 [1].

**Figure 2 animals-11-01800-f002:**
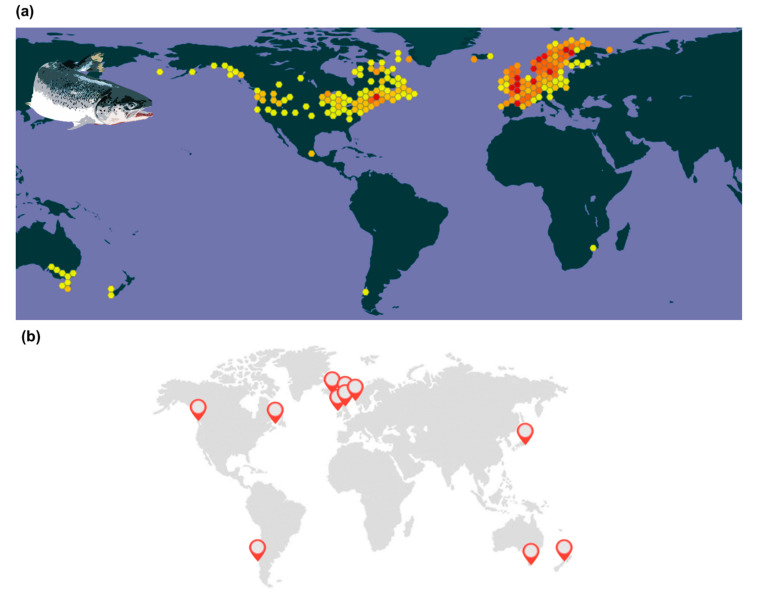
Distributional range of Atlantic salmon (*Salmo salar*). (**a**) Native and introduced with probabilities of occurrence ranging from yellow (lower) to red (higher) (GBIF Secretariat, 2020); and (**b**) main coastal areas suitable for salmon farming at specific latitudes with a thermal range between 0 to 18–20 °C (optimal water current conditions are commonly found in archipelagos and fjords) (adapted from [32]).

**Figure 3 animals-11-01800-f003:**
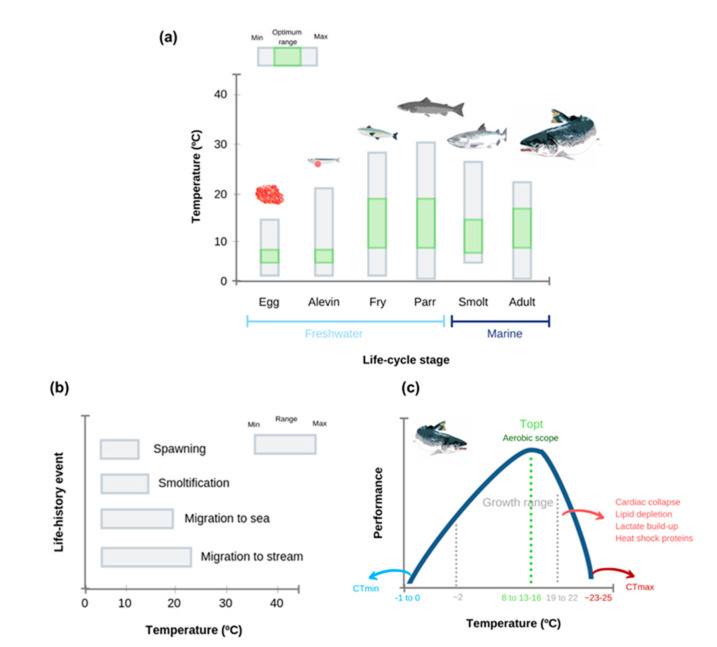
Thermal biology of Atlantic salmon (*Salmo salar*). (**a**) Thermal windows estimated along different life stages; (**b**) thermal windows for life-history events; and (**c**) hypothetical thermal performance curve for adult Atlantic salmon. Top—optimum temperature; CTmin—Critical Thermal Minimum; CTmax—Critical Thermal Maximum. Ranges were estimated from literature (see Appendix A for details). Please note that CTMax values may overestimate water temperatures that salmon can tolerate in culture and/or in the wild due to experimental protocols commonly employed (e.g., faster rates of water heating).

## Data Availability

Data sharing is not applicable to this article.

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
