# Peer review of "Summer Is Coming! Tackling Ocean Warming in Atlantic Salmon Cage Farming"

_animals, 2021, doi:10.3390/ani11061800_

Round 1
Reviewer 1 Report
General Comments:
I think this review has significant potential with regards to identifying challenges faced by Atlantic salmon at warm temperatures, and possible ways that the cage-site salmon aquaculture industry can adapt to climate change. The review is timely, and could make a significant contribution to the literature, and inform the industry on this topic. However: 1) there are several aspects of this paper that are misleading / where the literature on fish thermal biology is not properly interpreted; 2) there has been a lot of literature published on this topic recently (i.e., in the last two years), some of it very key/germane to the paper that is not included/cited; and 3) some other (additional) aspects of this topic should be considered / addressed in this review. Below I provide some specific comments/suggestions for revising the manuscript, and provide a number of references for the authors, some of this work only now being published. However, the authors need to become more knowledgeable in the area of fish thermal tolerance/biology and factors that need to be considered, including beyond the issues that I address, and the paper needs to be carefully edited. Addressing these issues will significantly improve the quality and impact of their review.
Specific Comments:
Line 15 and elsewhere….I think Atlantic salmon is now in the top 5 of produced fish species?
Line 48. Please change to 2 million metric tonnes (MMT)
The authors need to devote significant space/text (I would suggest a separate section) to talking about the sublethal effects of elevated temperatures on salmon physiology / biology. For example, at what temperatures are feeding and growth diminished? At what temperatures do the fish begin to experience stress etc. See Gamperl et al (2020), and the literature cited in this paper as a start.
The authors refer several times to the mass mortality event in Newfoundland that occurred in 2019. However, they refer only to newspaper articles, and fail to cite the report that was generated. Here is the link to the report (https://www.gov.nl.ca/ffa/files/publications-pdf-2019-salmon-review-final-report.pdf). This review cited high temperatures and hypoxia, along with sea lice treatments as the cause of death. However, recent research on these exact same stocks (see Gamperl et al., 2020) shows that no mortalities occured in these fish/stocks when given an incremental temperature increase (1oC per week, to simulate temperature changes at sea-cages in the North Atlantic in the spring/summer) until 21-22oC, even when this challenge was combined with exposure to oxygen levels of 60-70% air saturation. Further, several Norwegian studies, and one study in Australia, have also shown that salmon survive temperatures of 18-20oC for weeks with few/no ill effects, and Hvas et al. (2017) reported that Atlantic salmon can survive acclimation to 23oC for 2 weeks, and that mortality was only 20% after 4 weeks. So, it can be concluded that the salmon in NL did not die of the effects of hypoxia/temperature. As you can see in the report, they were also treating all sea-cages for sea lice. Sea lice treatments in themselves can lead to significant (15-30%; see Overton et al., 2019;
https://doi.org/10.1111/raq.12299) mortalities at sea-cage sites at normal temperatures, but these treatments at ~18oC were likely a significant contributor to the mortalities in NL. Further, a recent study (Godwin, 2020; see citation below) shows that ‘high’ levels of sea-lice infestation (6.8 ± 0.4 lice per fish) decreased the survival probability of post-smolt salmon at 19oC by 25%. So…it was a combination of sea lice infestation and high temperatures that was the issue…not temperature itself or hypoxia.
This is one area that the authors need to a better job...addressing the impacts of combined stressors….for example based on the existing data…salmon should be able to survive at 21-220C, combined with hypoxia down to ~ 65% air saturation with few mortalities….but abiotic-biotic interactions at these temperatures need to be carefully controlled etc…e.g. sea lice, disease (what are high temperature effects on fish immunology/disease resistance?
Most of the above references/information can be found in Gamperl et al. (2020; Aquaculture. 51. DOI: doi.org/10.1016/j.aquaculture.2019.734874), and the authors should be sure to cite/incorporate Stehfast et al., 2017; https://doi.org/10.1038/s41598-017-04806-2) into their paper. There are many places in this paper where the information in these papers needs also to be incorporated.
Godwin, S. C., Fast, M. D., Kuparinen, A., Medcalf, K. E., & Hutchings, J. A. (2020). Increasing temperatures accentuate negative fitness consequences of a marine parasite. Scientific Reports, 10, 18467. doi:10.1038/s41598-020-74948-3
Line 129….The temperature ranges that the authors cite are too low…maximum temperatures at sea cages in Atlantic Canada approach 19-20oC in the summer (also see Gollock et al., 2006), and 23oC in Tasmania (Stehfast et al., 2017).
The authors rely too much on CTMax data in their paper, and on data from wild salmonids. There is now a lot more relevant data on the thermotolerance of aquaculture/farm reared Atlantic salmon. Further, CTmax is a measure of the physiological limits of a species when exposed to an acute change in temperature…and in most circumstances these tests are run at rates of temperature increase (up to 18oC h-1) that are not close to being ecological (i.e., < 2oC hr-1), and certainly not relevant to the sea-cage situation/environment. They are good for examining what physiological processes limit thermal tolerance (more on this later), but their relevance outside of this is very questionable…..and so is the data presented in Figure 3A…if they are CTMax data (there is no information in the legend).
As shown best in Bjornsson et al. (2007), the general increase in temperature at sea cages in Norway and Iceland in the spring/summer, and the same is true for Atlantic Canada, is about approx. 1oC every 5-7 days. So, a temperature tolerance test (the incremental temperature maximum test, ITMax test) has recently been used/designed to examine how the thermal tolerance of fish exposed to temperature changes of this temporal scale compare to that of a CTMax test. The original study (Zanuzzo et al. 2019; https://doi.org/10.1016/j.cbpa.2019.03.020) showed that the ITMax of cod (1oC increase in temperature every 4 days) was approx. the same as their CTMax, but when family structure was taken into account, CTMax and ITMax were not related. With regards to salmon, if the authors compare the data of Leeuwis et al. (2019) and Penney et al. (2014), and Gamperl et al. (2020), the ITMax of Atlantic salmon is ~ 3-4oC lower than their CTMax (26oC vs. 22-23oC, respectively), and the latter data line up very well with the data of Hvas et al. (2017)…i.e. salmon can tolerate temperatures up to 22-23oC…even for days/weeks. So, I suggest that the authors remove the data they present on CTMax unless they are talking about rapid temperature changes, It just isn’t relevant, and re-write this section.
On page 5 they introduce the OCLTT theory/concept, and try to relate it to the salmon’s thermal tolerance. However, the authors fail to communicate/indicate how controversial this theory is (I provide references below) or what other processes/mechanisms are associated with upper thermal tolerance in fishes. Clearly, the data of Hvas et al. (2017), where the aerobic scope of salmon was equal to/or higher at 23oC (the temperature at which the fish died) vs. at 15 or 18oC argues against the relevance of this concept for salmon exposed to slowly increasing temperatures or acclimated to different temperatures. Clearly, aerobic scope/respiration is important to thermal tolerance in fishes (e.g. see Gerber et al., 2020 Sci. Rep. 10: 21636), but what specifically determines the temperature at which they die is not yet clear. Also see reference below.
Andreassen AH, Hall P, Khatibzadeh P, Jutfelt F, Kermen F. 2020 Neural dysfunction at the upper
thermal limit in the zebrafish. BioRxiv preprint. (doi:10.1101/2020.12.28.424529)
Ern R, Norin T, Gamperl AK, Esbaugh AJ. 2016. Oxygen dependence of upper thermal limits in fishes. J. Exp. Biol. 219, 3376–3383. (doi:10.1242/jeb.143495)
Haverinen J, Vornanen M. 2020. Reduced ventricular excitability causes atrioventricular block and depression of heart rate in fish at critically high temperatures. J. Exp. Biol. 223, jeb225227. (doi:10.1242/jeb.225227)
Ern, R. (2019). A mechanistic oxygen- and temperature-limited metabolic niche framework. Philosophical Transactions of the Royal Society B: Biological Sciences, 374(1778), 6–10. https://doi.org/10.1098/rstb.2018.0540
Jutfelt, F. et al. (2018). Oxygen- and capacity-limited thermal tolerance: Blurring ecology and physiology. J. Exp. Biol. 221, 2016–2019
Timothy D. Clark, Erik Sandblom, Fredrik Jutfelt (2013). Aerobic scope measurements of fishes in an era of climate change: respirometry, relevance and recommendations J. Exp. Biol. doi: 10.1242/jeb.084251
Lines 219-225…this experiment/these data are not at all relevant given the thermal regime used to acquire this data. Delete…..but see Gamperl et al. 2020 for appropriate references/information with regards to salmon behavior at high temperatures.
Section 4 leaves out / does not address a number of important strategies/approaches that could be used to improve the thermal performance/tolerance of farmed Atlantic salmon. There is considerable information in these areas. Some of these areas include:
- Adjustments in nutrition. For example, Skretting has a high temperature (HT) diet that is used in Tasmania, and there are several studies that have looked at modifying nutrition in fish to improve thermal tolerance.
- There are several other studies that have used genomic/genetic approaches to examine what genes could be used to select fish with improved thermal tolerance. I have included some references below, but this section needs to be expanded considerably. If one reads a number of the genomics papers, there is a concensus with regards to particular genes, where specific efforts to identify different alleles/paralogues should be investigated.
Beemelmanns, F.S. Zanuzzo, R.M. Sandrelli, M.L. Rise and A.K. Gamperl (in press). The Atlantic salmon’s stress- and immune-related transcriptional responses to moderate hypoxia, an incremental temperature increase, and these challenges combined. G3: Genes. Genomics. and Genetics.
Beemelmanns, L. Ribas, D. Anastasiadi, J. Moraleda-Prados, F. S. Zanuzzo, M.L. Rise. and A.K Gamperl (2020). DNA methylation dynamics in Atlantic salmon (Salmo salar) challenged with high temperature and moderate hypoxia. Front. Mar. Sci. 7: Art. #604878. 10.3389/fmars.2020.604878
Beemelmanns; F.S. Zanuzzo; X. Xue; R.M. Sandrelli. M.L. Rise and A.K. Gamperl (2021). The Transcriptomic Responses of Atlantic Salmon (Salmo salar) to High Temperature Stress Alone, and in Combination with Moderate Hypoxia. BMC Genomics. DOI:10.21203/rs.3.rs-38228/v1
Nicole L. Quinn, Colin R. McGowan, Glenn A. Cooper, Ben F. Koop, and William S. Davidson (2011) Identification of genes associated with heat tolerance in Arctic charr exposed to acute thermal stress. https://doi.org/10.1152/physiolgenomics.00008.2011
Jeffries KM, Hinch SG, Sierocinski T, Clark TD, Eliason EJ, Donaldson MR, et al. (2012). Consequences of high temperatures and premature mortality on the transcriptome and blood physiology of wild adult sockeye salmon (Oncorhynchus nerka). Ecol Evol. 2:1747–64.
Olsvik PA, Vikesa V, Lie KK, Hevroy EM. Transcriptional responses to temperature and low oxygen stress in Atlantic salmon studied with next generation sequencing technology. BMC Genomics. 2013;14:817.
Houde et al. (2019) Salmonid gene expression biomarkers indicative of physiological responses to changes in salinity and temperature, but not dissolved oxygen. J. Exp. BIol.. 222: Article Number: jeb198036
….see several papers by Roy Danzmann
- Changes in rearing incubation / rearing temperature (i.e., possibly taking advantage of ‘developmental plasticity’) also have promise for improving the thermal tolerance of fish, including salmon…although much of this data is based on CTMax
Beaman, J. E., White, C. R., & Seebacher, F. (2016). Evolution of plasticity: mechanistic link between development and reversible acclimation. Trends in Ecology and Evolution, 31(3), 237–249. https://doi.org/10.1016/j.tree.2016.01.004
Del Rio, A. M., Davis, B. E., Fangue, N. A., & Todgham, A. E. (2019). Combined effects of warming and hypoxia on early life stage Chinook salmon physiology and development. Conservation Physiology, 7(1), 1–14. https://doi.org/10.1093/conphys/coy078
Schaefer, J., & Ryan, A. (2006). Developmental plasticity in the thermal tolerance of zebrafish Danio rerio. Journal of Fish Biology, 69(3), 722–734. https://doi.org/10.1111/j.1095-8649.2006.01145.x
Figure 4 is not particularly useful, and could/should be deleted.
Author Response
Dear Reviewer 1, please see the uploaded file entitled "Reply to reviewers comments animals-1160351".

Reviewer 2 Report
The authors have reviewed the effect of global warming on salmon marine aquaculture. This is an interesting topic that has filled the gap in our knowledge about the interactions between ocean warming and marine aquaculture. The manuscript (MS) was well-written, and most of the parts were easy to understand. They covered most of the topics that were needed. However, some issues have compromised the quality of this MS and need to be addressed. I did not reject this MS to provide a chance for authors to revise the MS. However, a lot of efforts are required to revise this MS and improve it. We all know that climate change is here right now, but it should not be too focused, especially in ocean farming.
Major comments
- One issue that caused misconception in most parts of this MS is that the authors did not mention the temperature in the cited studies. Most of the references are talking about temperatures upper than 25 degrees that do not make sense in reality for salmon farms. To solve this, I suggest authors clearly mention temperature on any single reference that they cite. Further, you can delete some laboratory studies that tested too high temperature or thermal stress (like upper than 25-26 degrees for a short period).
- Another challenging issue that causes many misconceptions here is that the authors overestimated thermal stress in some places. In the north, in Chile and Norway, heat stress has not been any problem and even has been useful for salmon farms. In the south, for example, Australia (produce less than 5% of salmon production), there is another story, and this question should be asked whether Australia is a good place for farming salmon or not!!!!. Therefore, authors should mention the place of references that they cite to avoid misconceptions.
- The increase of ocean temperature would be 1 degree for like 50 years, which will not make any problem for salmon farms. This issue was also overestimated too much in this MS.
- Further, 90% of salmon production is for areas that are so unlikely the temperature goes upper to 18 degrees. From my idea, the problem of less than 10% of salmon farms has been generalized for all 100% salmon aquaculture that has caused misconceptions. The thing you can do for this is that you can provide more details about the place, temperature, and the times that temperature was upper than optimum level.
- I will review the MS again after the authors could revise the MS and decrease the misconceptions.
Minor comments:
Abstract
- Line 12-13, I suggest deleting this part, and no one knows it is true or wrong.
- Line 15-20, please explain this part in the MS with references. I do not think the heat wave can kill salmon, and even in some parts like Norway has been useful, and the heatwave closed the temperature to optimum for farming salmons. I suggest revising this part.
- Line 12-24, please mention somewhere here what temperature is “high” for salmonids.
- Line 27, add temperature
- Line 27-29, please be clearer how often these waves are and how much temperature will be increased.
- Line 39-40, I just think authors overestimated the effect of ocean warming as it has been less than 1 degree in many parts.
Introduction:
- Line 57-59, my major comment that I already mentioned. However, the heat wave that authors were reported was just one expectation (118 days upper than 18 degrees!!). If you look climate data up in Tasmania, you can usually see this number is less than 30-40 days during the year. The point that I wanted to make is that in this MS, be clear about the places and when you cite a reference, please provide details about the place, how much was temperature, how many days were exposed to high temperature. Also, 90% of salmon production is for areas that never temperature goes upper than 17 degrees. The thing you can do is that, as I said above, provide more details about the place, temperature, and the times that are upper than optimum level.
- Line 65-66, I do not agree with this; please mention how much temperature has been increased.
- Line 84-86, You have already pointed my concerns here. There is just 1 degree for 40 years, and there is no problem for 90% of salmon farms. And would be like 1-2 degrees in the next century.
- Please increase the quality of pictures as the requirements of the journals were not provided
- You mentioned my other concern in Figure1. Less than 5% of salmon farms can be exposed to high temperatures or heat waves.
- Line 130, please double-check the references; they are not present the point that you made. I wish the temperature was 16-18 degrees in summer in Russia and Norway!!!. Please double-check and use the climate data which is available for the public. By looking at this data, you would see the problem has been overestimated in this MS.
- Line 38, this is just a prediction for natural resources of Atlantic salmon and not aquaculture farms. Please revise the MS from this point to avoid any misconception. I suggest checking any single reference that you used to make sure it is fitted for this MS and ideas.
- Line 193-187, Another kind of misconception happened here. Salmon cannot be farmed upper to 18-19 degrees at the moment. Therefore, all these things you mentioned do not make sense in reality. I highlighted this point again: less than 5% of salmon farms have problems related to high temperature during some days in summer.
- Line 189, please mention to temperature
- Line 213, please double check these references, there are modeling, or the rate of negative impact is too low.
- Line 218, please provide more details about the places and how often the wave has occurred.
- I did not review the rest of the MS as it needs first to be massively revised from the points that I already mentioned.
Best regards
Good luck with the revision, and I hope I have contributed to improving this MS.
Author Response
Dear Reviewer 2, please see the uploaded file entitled "Reply to reviewers comments animals-1160351".

Reviewer 3 Report
Review
Paper title: Summer is coming! Tackling ocean warming in Atlantic salmon cage farming.
Due to the high quality of its meat, Atlantic salmon has become a common and valuable object for aquaculture. Salmon farms are traditionally located at high latitudes because of the thermal requirements of Salmo salar. Recent Ocean warming has raised a problem for finfish aquaculture of cold-water species and reliable solutions are required to fix this concern. The authors summarized the data regarding thermal tolerance limits of Atlantic salmon and the negative consequences of heat stress. We authors provided possible ways to minimize negative effects in this industry.
All these reasons explain the relevance of the paper by R. Calado and co-authors submitted to "Animals".
General scores.
The data presented by the authors are significant. The authors considered and comprehended relevant literature sources to provide an excellent overview in this field. We authors conducted careful work which will attract the attention of a wide range of specialists including aquaculturists, stakeholders, and managers focused on cage farming.
I recommend this paper for publication after minor revisions.
Specific comments.
L 14. Change “well develop” to “well developed”
L 21-23. Change “more advance” to “more advanced”
L 48. Change “2 M tons” to “2 million tons”
L 49. Change “a reduced short-run elasticity” to “reduced short-run elasticity”
L 54. Change “Other factor” to “Other factors”
L 120. The authors use both “tons” and “tonnes”. They should be consistent throughout the text.
L 124. Change “mortality” to “and mortality”
L 179. Change “Corey, et al.” to “Corey et al.”
L 219. Change “regards animal welfare” to “regards to animal welfare”
L 230. Change “impact in” to “impact on”
L 233. Change “impact its” to “impact their”
L 288. Change “Imsland, et al.” to “Imsland et al.”
L 305. Change “in USD 5.5 million” to “at $5.5 million”
L 307. Change “millions of Atlantic” to “millions Atlantic”
L 308. Change “USD 82 million” to “$82 million”
L 315. Change “seacage” to “sea cage”, “but has” to “but have”
L 328. “these shift”. “this shifts [what?]”
L 345. Change “the northern” to “northern”
L 358. Change “virulence associated” to “virulence-associated”
L 361. Change “outbreak-associate” to “outbreak-associated”
L 405. Change “sand” to “and”
L 406. Change “climate change maintain” to “climate change maintains”
L 407. Change “a refugia” to “a refugium”
L 425. Change “by decreasing” to “decreasing”
L 444. Change “Glover, et al.” to “Glover et al.”
L 445. Change “Solberg, et al.” to “Solberg et al.”, “bodyweight” to “body weight”
L 446. Change “bodyweight” to “body weight”
L 450. Change “Sae-Lim, et al.” to “Sae-Lim et al.”
L 459. Change “detailed on” to “detailed in”
L 503. Change “role on” to “role in”
L 517. Change “Yang, et al.” to “Yang et al.”
L 527. Change “contribute to pave” to “contribute to paving”
L 539-540. Delete “Please turn to the CRediT taxonomy for the term explanation. Authorship must be limited to those who have contributed substantially to the work reported.”
L 622. Missing Publisher details
L 623. Missing Publisher details
L 625. Missing Publisher details
L 629. “Salmo salar” should be italicized.
L 634. “Salmo salar” should be italicized.
L 639. “Oncorhynchus” should be italicized.
L 649. “Salmo salar” should be italicized.
L 657. “Salmo salar” should be italicized.
L 664. “Salmo salar” should be italicized.
L 669. “Salmo salar” should be italicized.
L 672. “Salmo salar” should be italicized.
L 675. “Salmo salar” should be italicized.
L 678. “Salmo salar” should be italicized.
L 687. “Salmo salar” should be italicized. “Salmo trutta” should be italicized
L 688. “Salvelinus alpinus” should be italicized.
L 696. “Salmo salar” should be italicized.
L 717. “Salmo salar” should be italicized.
L 719. “Salmo salar” should be italicized.
L 724. “Salmo salar” should be italicized.
L 728. Missing link.
L 739. “Salmo salar” should be italicized.
L 745. “Salmo salar” should be italicized.
L 751. Delete “BMC Evolutionary Biology”.
L 752. “Salmo salar” should be italicized.
L 764. “Salmo salar” should be italicized.
L 801. “Salmo salar” should be italicized.
L 809. Change “Ices” to “ICES”
L 815. “Salmo salar” should be italicized.
L 829. “Salmo salar” should be italicized.
Figures 1-4 have too low resolution to be published in the Journal.
Author Response
Dear Reviewer 3, please see the uploaded file entitled "Reply to reviewers comments animals-1160351".

Round 2
Reviewer 1 Report
The authors have made a number of substantial revisions to this paper/manuscript, and it is much improved. However, there are a number of other things that should be considered/addressed, and it really needs a careful editing. I have pointed out a number of issues, but there are many spelling mistakes and issues with tense/grammar that also need attention. If possible, I would suggest that they have a colleague whose first language is English carefully go through the manuscript.
Lines 18 and 29: Statement is not true...rarely (except in Tasmania) are SW temperatures above values tolerated by the species. Please revise.
Line 19…Please replace ‘promote’ with ‘can be associate with’
Lines 20 and 31…should be ‘This paper reviews’…
Lines 23 and 36. Replace ‘on with ‘in’
Lines 38 to 41. These are not the only possible solutions.
Line 45. In the introduction the authors need to acknowledge that there have been more losses at salmon cage-sites due to low temperatures that warm water events over the past decade, and these events are also expected to get worse with climate change…but, this review will focus on warm water temperatures.
Szekeres, P., Eliason, E., Lapointe, D., Donaldson, M., Brownscombe, J., & Cooke, S. (2016). On the neglected cold side of climate change and what it means to fish. Climate Research, 69(3), 239–245. https://doi.org/10.3354/cr01404
Line 48…delete the ‘M’
Line 55. ‘factord’ should be ‘factors’
Lines 55 – 58. This is a very awkward sentence. Please re-write/revise, and consider breaking into 2 sentences.
Line 68. The data clearly shows that sea-water temperatures were not the issue, they were at least 2-3oC below what these stocks can tolerate…you need to be very careful about your interpretation here…other issues resulted in mortalities at normally sub-lethal temperatures. Please revise here and elsewhere in the document.
Lines 74-75. What do you mean by ‘due to therapeutics’?
Lines 84-85…again you are only addressing upper temperature limits in this review….be specific for the reader.
Line 170. Please replace ‘enhanced’ with ‘increased’
Line 172…’Arctic’ needs to be capitalized.
Line 181. Why would the Atlantic salmon be susceptible to precipitation? It is an anadromous species, and can be reared in brackish water. Please clarify.
Line 210. Remove bracket after ‘introduced’.
Line 222. Should be ‘acute exposure to high temperatures’. Also, the rest of this sentence has several issues and needs to be re-written.
Line 222-224. Why present data for alevins/fry, when this article is about life history stages that are put into the sea-cages. They are reared in hatcheries where temperatures are controlled or cool. For cardiovascular data on fish of this size/life-stage see articles Penney et al. (2014) and Leeuwis et al. (2019).
Lines 226 – 228. Again….the upper thermal maximum of ‘parr’ is not relevant to this article. Also, the rate of heating provided data that are non-physiological or relevant to fish in streams, the oceans, or sea- cages. This data is very misleading, and such data should not be included in this review. Delete. Please revise/shorten the section between 222 and 235.
Lines 247-249. Not according to Hvas et al. (2017).
Lines 251-253. The comprehensive data sets presented by Beemelmann’s et al. (2020; 2021a and b) show that changes in stress protein transcript expression start by 18oC. Please refine this description. Here is an additional citation for you (DOI: 10.1093/g3journal/jkab102)
Line 273. There is no need to capitalize ‘reactive oxygen species’.
Line 279. These authors do not suggest that this is the only mechanism…only that this mechanism was involved (at least correlated) with changes in the expression of specific genes. So ‘partially mediated’ might/would be more appropriate.
Lines 314-316. Water temperatures in the winter in Iceland and Atlantic Canada regularly get down to 0-1oC, and thus, this sentence needs to be revised.
Line 345 and elsewhere in the manuscript…should be ‘salmon’ not ‘salmons’
Lines 348-355. No references are provided for this section.
Lines 356- 358. Please revise the grammar, structure, in this sentence.
Line 360. Should be ‘on’ not ‘ion’
Line 363. Delete ‘their’
Figure 3 caption. Please put a ‘cautionary statement in here’ with respect to panel (a)….The reported CTMax values may overestimate what salmon can tolerate in culture / in the wild, due to aspects of the protocol (i.e., the very rapid rate of heating). Please also consider this comment with regards to supplementary Table 1.
Lines 432-434. This is the exact same sentence as in lines 317-319?
Line 443. This information is somewhat misleading. In the two studies in Gamperl et al. (2020), the salmon were ~ 400 and 800 g when they reached ~20oC….
Lines 447-448. Please correct sentence structure/grammar issues.
Line 489. ‘River’ should also be capitalized.
Line 493. Should be ‘have’
Line 507. ‘theseis shifts according’?
Line 517. Should be ‘ 133])’
Line 545. ‘mean size at sea age’? What does this mean?’
Line 560. I think it is very important to indicate the range in CTMax between these salmon families…it was at maximum ~ 2.0oC….this agrees with a study on Atlantic cod recently published by Zanuzzo et al. (2019). So, this research suggests that the capacity for improvement of upper thermal tolerance might be limited. What do other studies on salmonids say/suggest about what increase in thermal tolerance might be achievable with breeding etc. What do studies on redband trout and rainbow trout in Western Australia etc. tell us (see papers by Chen et al. 2015 and 2018? This comment also has relevance to the section from lines 639-657.
Line 578. Delete ‘)’ after 20oC
Line 583. Salmonids do not have a larval stage…egg, eleuthroembryo/alevin, fry, parr……
Lines 596-627. The title of this section is ‘4. Addressing temperature-related stress in salmon farming using the biotechnology toolbox’. This text has nothing to do with this topic, and is related to the susceptibility of wild populations to climate change. This, I suggest that you provide a new title.
Line 606. ‘refugiuma”?
Line 623. ‘Should be ‘pools’ not polls’
Line 636. Should be ‘selectively breed’
Line 674. ‘Bby’
Line 678 ‘ion’
Line 702. Should be ‘paving’
Author Response
Please see the attached file with our Reply to Reviewers Comments Second Round of Reviews

Reviewer 2 Report
The authors have improved the quality of this MS, but some more works are required to get it close to the final version.
- General comment: I suggesting reduce the text of this MS to at least 30%. Most of the parts needed to be summarized, and some less-related sentences can be deleted. Your MS has 600 line text!! Which is better to be a maximum of 450 lines.
- Line 38, please check whether the “gene editing” can be applicable for thermal tolerance or not. If not, I suggest changing it to “omics technologies.”
- Keywords: please capitalize each word and order them alphabetically.
- Line 45, change to “factors,”
- Line 91, please summarize this section and reduce it by at least 30%.
- Line 222, correct the error of “s”
- Generally, please revise the MS to correct some writing errors and delete extra spaces.
- Line 223 and elsewhere, please be consistent with a common name, and for the first time in MS, please report common name plus scientific name and rest just common name.
- Line 223-311, I suggest summarizing this part.
- I am not the optimum temperature (~19-22 ËšC) for Atlantic salmon, to be correct. Please double-check it. (https://doi.org/10.1016/j.aquaculture.2008.06.042 ; and many other studies.
- Line 244 The thermal tolerance polygon….until the end of the page, please summarize this section.
- Line 280, high temperature (?), add the number.
- Page 7, In particular… here and elsewhere, please make sure you used up-regulated and down-regulated correctly. Please check the difference between these phrases with overexpression and etc. I think the correct phrase is not “upregulation” in this study. Please check this point for other cited studies as well.
- One general point: You have discussed in this review “cage farming” and marine farm, which is about adult salmon. Therefore, some explanation from page 7 and other parts related to freshwater (juvenile stage) is not fitted to this MS. I suggest deleting them and revising the MS from this point and only focus on studies on adult age in marine water.
- Page 7, line 312 … Exactly, this is optimum temperature and not (19-22). Please review the MS and make sure you suggest the one reasonable suggestion for optimum temperature for adult that would be (17-19 ËšC) according to the hips of papers.
- Line 360, please revise it.
- Line 430-488, please make sure you only focus on marine waters studies. Further, this section needs to be summarized; I suggest focusing only on salmon studies as enough references are available.
- Line 500, please summarize this section and reduce it by at least 30%.
- Line 502-529, please delete this part and add a sentence to start this section and connect to the following details related to phenotypic plasticity.
- For section 5, preferably delete the studies that are in freshwater. If marine studies are not enough for this section, please clarify when you cite a freshwater study.
- Line 639, please revise it as it is not clear enough.
- Line 678, change to Genetic engineering and omics technologies.
- Line 677, please summarize it at least 30%, too much information was provided.
Best regards
Author Response
Please see the atatched file with our Reply to Reviewers Comments Second Round of Reviews
